# Do Women Have a Higher Mortality Risk Than Men following ICU Admission after Out-of-Hospital Cardiac Arrest? A Retrospective Cohort Analysis

**DOI:** 10.3390/jcm10184286

**Published:** 2021-09-21

**Authors:** Christiaan van Wees, Wim Rietdijk, Loes Mandigers, Marisa van der Graaf, Niels T. B. Scholte, Karst O. Adriaansens, Remco C. M. van den Berg, Corstiaan A. den Uil

**Affiliations:** 1Department of Intensive Care, Erasmus University Medical Center, 3015 GD Rotterdam, The Netherlands; c.vanwees@erasmusmc.nl (C.v.W.); L.mandigers@erasmusmc.nl (L.M.); c.denuil@erasmusmc.nl (C.A.d.U.); 2Department of Cardiology, Erasmus University Medical Center, 3015 GD Rotterdam, The Netherlands; marisavdgraaf@outlook.com (M.v.d.G.); n.scholte@erasmusmc.nl (N.T.B.S.); karst.adriaansens@gmail.com (K.O.A.); 3Department of Hospital Pharmacy, Erasmus University Medical Center, Doctor Molewaterplein 40, 3015 GD Rotterdam, The Netherlands; 4Department of Intensive Care, Haaglanden Medical Center, 2512 VA The Hague, The Netherlands; remco.vd.berg@hotmail.com; 5Department of Intensive Care, Maasstad Ziekenhuis, 3079 DZ Rotterdam, The Netherlands

**Keywords:** cardiac arrest, intensive care unit, sex differences, 90-day mortality, out-of-hospital

## Abstract

Purpose: previous studies showed that women have a higher mortality risk than men after out-of-hospital cardiac arrest (OHCA). This sex difference may disappear after adjustment for cardiac arrest characteristics. Most studies also included patients who were not admitted to the intensive care unit (ICU). We analyzed whether sex impacts the mortality of ICU-admitted OHCA patients. Methods: a retrospective cohort analysis of 1240 OHCA patients admitted to the ICU (310 women, 25%, Age_Median_ 64.0 (IQR 53.8–73.0)) at an academic hospital in the Netherlands between 1 January 2007 and 31 December 2018. The primary outcome was 90-day mortality; the secondary outcome was a favorable cerebral performance category (CPC) score at ICU discharge and ICU length of stay (ICU LOS). Results: we found no association between sex and 90-day mortality (hazard ratio (HR) 0.867; 95% confidence interval (95% CI) 0.678–1.108) after adjusting for relevant cardiac arrest characteristics. Similarly, we found no difference for favorable CPC score (OR 1.117; 95% CI 0.777–1.608) or ICU LOS between sexes (Beta 0.428; 95% CI −0.442 to 1.298). Conclusions: after adjusting for cardiac arrest characteristics, we found no difference between women and men with respect to 90-day mortality, ICU LOS, and CPC score.

## 1. Background

Out-of-hospital cardiac arrest (OHCA) is considered a worldwide problem and the leading cause of death in Europe and the United States [1,2]. The incidence of OHCA in the Netherlands is 62 cases per 100,000 person–years annually, of which, 47 underwent cardiopulmonary resuscitation (CPR) [1,3]. A meta-analysis reported an overall survival of 7% in adults after OHCA and a successful resuscitation to discharge rate of 6–31% in the Netherlands [4].

In recent years, progression in rapid provision of basic life support (BLS) and defibrillation have increased a patient’s chance of survival [5]. Systems, such as the use of text alerted lay rescuers, education of medical personnel, and the distribution of public access automated external defibrillators (AEDs), have all proven to increase survival chances [6].

Even with these developments in pre-hospital care, theoretically impacting the survival of both sexes, there is still some difference found in mortality after OHCA between women and men. Several studies investigated the factors that could explain why there may be a sex difference [7,8]. Women may be less likely to receive lay rescuer BLS, an AED may be less frequently used, they are older, and have shockable rhythms less often than men [8]. These differences may be partially explained by the fact that coronary artery disease symptoms that precede an OHCA may present differently in women [9,10].

Previous studies analyzed sex differences after OHCA, but mixed results were found [11]. Some investigators reported a worse outcome for women after OHCA [8,12]. However other studies suggested no difference or even the opposite [13,14]. Using a nationwide database, our research group recently found a higher 1-year mortality in women admitted to the intensive care units (ICUs) after OHCA and IHCA compared to men [15]. However, in this study some baseline characteristics, mostly regarding cardiac arrest parameters (e.g., initial cardiac rhythm, cause of arrest), were unknown, hampering ideal adjustment for confounders.

For this reason, our aim was to examine whether sex differences remain significant after adjustment for OHCA characteristics. Therefore, we analyzed this sex difference in a cohort of OHCA patients admitted to the ICU in an academic hospital in the Netherlands. Studying the outcomes of these critically ill CA patients is highly interesting because these patients survive the first episode of CA (namely, cardiopulmonary resuscitation), while they are still prone to hemodynamic deterioration/instability, ischemia/reperfusion injury, and neurological damage [15].

## 2. Material and Methods

### 2.1. Study Sample, Setting, and Design

We performed a retrospective cohort study of adult OHCA patients who survived until admission to the ICU, at the Erasmus University Medical Center, Rotterdam, the Netherlands, between January 1st, 2007 and December 31st, 2018. The data were extracted from the electronic patient record and stored in a secured database that is open for reuse. Data extraction was conducted by several authors (i.e., C.v.W., M.v.d.G., K.A., and R.v.d.B.). The extracted data consisted of patient characteristics, such as demographic characteristics, medical history, medication use history, cardiac arrest characteristics, and clinical outcomes. We included adult (≥18 years) patients of whom arrest characteristics and the ICU stay was known. The medical ethics committee of the Erasmus MC approved the study, and the need for informed consent was waived (MEC-2019-0206).

### 2.2. Baseline Characteristics and Clinical Outcomes

We collected several baseline characteristics and clinical outcomes of these patients. First, as demographic characteristics, we collected age and gender. Second, as medical history, we collected history on hypertension, hypercholesterolemia, diabetes, family history of cardiovascular disease, former myocardial infarction, chronic heart failure, cardiac arrhythmias, previous transient ischemic attack or stroke, peripheral vascular disease, smoking, chronic lung disease (e.g., asthma and COPD), pulmonary embolism, previous coronary artery bypass grafting, previous percutaneous coronary intervention, and previous internal cardioverter defibrillator (ICD) or pacemaker implantation. Third, on medication history, we collected data on the use of aspirin, antiplatelet therapy, oral anticoagulants, ACE inhibitors, aldosterone antagonists, angiotensin-II blockers, beta blockers, calcium channel blockers, nitrates, diuretics, statins, non-statin lipid lowering agents, and antiarrhythmic. Fourth, as cardiac arrest characteristics, we collected location of arrest, witnessed arrest, bystander CPR, estimated time until the start of CPR, defibrillation by AED, initial rhythm, defibrillation by emergency medical service (EMS), time interval until ROSC, out of hospital intubation, Glasgow coma scale (GCS) score at emergency department (ED) and ICU admission, out of hospital sedation, and cause of arrest. Fifth, as for clinical outcomes, we collected ICU length of stay, cerebral performance category (CPC) score at ICU discharge, and mortality status at 90-day (i.e., 90-day mortality).

### 2.3. Primary and Secondary Outcomes

The primary outcome of this study is 90-day mortality after ICU admission. The secondary outcome was neurological status at ICU discharge, defined as a favorable CPC score. The CPC ranged from 1 to 5. The CPC score was dichotomized grouping scores 1 (= good cerebral function/minor disability) and 2 (= moderate disability) as favorable neurological outcomes. Contrary, scores 3 (= severe disability), 4 (= vegetative state), and 5 (= dead) were considered unfavorable neurological outcomes. The CPC score was based on the discharge report of the attending neurologist. This scoring was conducted by several authors (i.e., C.v.W., M.v.d.G., K.A., and R.v.d.B.) managing the database and used in a recent study by Mandigers et al. [16]. The other secondary outcome was ICU length of stay.

### 2.4. Statistical Analysis

We analyzed the data in three steps. First, descriptive statistics of the baseline characteristics are reported as number (percentage, %) or median (interquartile range, IQR) for categorical or continuous variables, respectively. To test for differences between 90-day non-survivors and survivors, and women and men, we used Chi-square tests for categorical variables and the independent sample t-test for continuous variables. At this step, we analyzed the data using pair-wise deletion in order to get the most information available from the data. Where there are missing values, we will present the numbers per variable separately.

To study the primary outcome, 90-day mortality, we performed multivariable Cox proportional hazard regression comparing women and men. We adjusted this model for baseline characteristics that were clinically relevant based on previous studies and were significantly different between women and men in the baseline characteristics. The result of the Cox proportional hazard regression (‘Cox regression’) is presented as hazard ratio (HR) and 95% confidence interval (95% CI). For illustrative purposes, we presented a Kaplan–Meier curve (and the associated log-rank test) comparing 90-day mortality for women and men.

To analyze the secondary outcomes, favorable CPC score and ICU length of stay, we performed a binary logistic regression and a linear regression comparing women and men, respectively. These results are presented as odds ratio (OR), Beta, and 95% CI. As for ICU LOS, we performed a linear regression for the association between sex and ICU LOS comparing women and men, respectively. Again, both the binary logistic and linear regressions were adjusted for the same baseline and cardiac arrest characteristics as the Cox regression.

As a sensitivity analysis, we analyzed the association between sex and ICU LOS and favorable CPC score for hospital survivors. The results of these regressions—using a similar set-up as the main secondary outcome analysis—are presented in Appendix A. The regressions were based on list-wise deletion, i.e., including only full patient data. This results in a somewhat lower sample size. In order to control for the effect of a smaller sample and to check the consistency of the regression results, we performed a bootstrapping procedure as available in SPSS for all three regression models (using 1000 samples). All analyses were performed using SPSS version 26 (Armonk, NY, USA: IBM corp) and *p*-values < 0.05 were considered statistically significant.

## 3. Results

### 3.1. Study Sample

Initially, 1524 patients were in the database, of whom 284 patients were excluded as they were below 18 years (N = 3), ICU data were missing (N = 35), and location of arrest was unknown (N = 246). Eventually, we included 1240 (310 women and 930 men) patients in this study (Figure 1).

### 3.2. Descriptive Statistics: Baseline Characteristics

The descriptive statistics are shown in Table 1 comparing non-survivors and survivors at 90 days after an OHCA. We found a near statistically significant difference in sex at 90-day mortality (*p* = 0.053). Patients who died before 90 days had a higher median age compared to survivors at 90 days (68.5 years vs. 60.7, *p* < 0.001). Further, we found that OHCA patients resuscitated at home had a higher likelihood of 90-day mortality (62.2% versus 37.8% for OHCA that occurred in the public space, *p* < 0.001).

Descriptive statistics comparing men and women are presented in Table 2. This table shows that men have significantly higher median age (64.4 vs. 61.8 years, *p* = 0.014), are more likely to have had percutaneous coronary intervention (PCI; 13.5% vs. 8.1%, *p* = 0.001), or coronary artery bypass grafting (CABG; 6.6% vs. 2.9%, *p* = 0.003). Women are also less likely to have an initial shockable rhythm (67.2% vs. 82.5%, *p* < 0.001), AED defibrillation (29.5%, vs. 38.2%, *p* = 0.006), and defibrillation by EMS (57.0% vs. 66.5%, *p* = 0.003). We found that in women the arrest occurs more frequently at home than in men (68.1% vs. 47.6%, *p* < 0.001). Finally, it is important to note that for smoking and GCS at the ED admission (GCS at ED), we report the descriptive statistics, but due to many missing values, these variables are not included in the multivariable analysis.

Table 3 presents the clinical outcome comparing women and men. We found that women have no statistically significantly different 90-day mortality risk than men (45.8% vs. 40.0%, *p* = 0.053). Further, we found no difference between women and men for ICU LOS (4 vs. 4 days, *p* = 0.084). Last, a favorable CPC score at ICU discharge is significantly different comparing women and men (50.0% vs. 57.0%, *p* = 0.038).

### 3.3. Primary Outcome: 90-Day Mortality

Figure 2 shows a Kaplan–Meier curve comparing the time to the 90-day mortality for women and men separately. The 90-day mortality was significantly higher in women than in men (log-rank test *p* = 0.029). We included sex, age, hypercholesterolemia, history of myocardial infarction, chronic lung disease, previous pulmonary embolism, previous PCI or CABG, aspirin use, calcium channel blocker use, statin use, location of arrest, AED defibrillation, initial rhythm, defibrillation by EMS, and cause of arrest in our multivariable Cox regression model. Table 4 shows that in the adjusted multivariable Cox regression model. The observed difference between women and men, regarding the time to the 90-day mortality, was not statistically significant (HR 0.870, 95% CI 0.679–1.114).

In addition, we found several other important baseline characteristics were associated with the time to 90-day mortality in the Cox regression analysis. First, we found that AED use was associated with a lower 90-day mortality (HR 0.540, 95% CI 0.414–0.705). Second, initial rhythm (i.e., shockable vs. non-shockable) was associated with a lower time to 90-day mortality (shockable HR 0.380, 95% CI 0.287–0.505). Third, a home arrest compared to a public arrest was positively associated with time to 90-day mortality (HR 1.606, 95% CI 1.293–1.996). After bootstrapping, we drew similar conclusions as the main analysis presented here. The bootstrapping results are available upon request from the corresponding author.

### 3.4. Secondary Outcome: CPC and ICU Length of Stay

Table 5 shows the linear and binary logistic regression analyses for ICU length of stay and favorable CPC score at ICU discharge, respectively. As for ICU length of stay, we found no difference between women and men (Beta 0.428; 95% CI −0.442 to 1.298). As for favorable CPC score, we found no statistically significant difference comparing women and men (OR 1.123, 95% CI 0.780–1.617). In addition, we found that AED defibrillation has a higher likelihood of a favorable CPC score at ICU discharge (OR 2.071, 95% CI 1.457–2.945). After bootstrapping, we drew similar conclusions as the main analysis. The bootstrapping results are available on request from the corresponding author.

### 3.5. Sensitivity Analysis

As a sensitivity analysis, we repeated our secondary outcome analysis in ICU survivors. The results are presented in Appendix A. We found that there is still no difference between women and men and their ICU LOS (Beta 0.685, 95% CI −0.505–1.875) and favorable CPC score at ICU discharge (OR 1.341, 95% CI 0.654–2.746).

## 4. Discussion

The aim of this study was to examine whether sex differences remain statistically significant after controlling for cardiac arrest characteristics in a sample of OHCA patients. Our results showed no difference in 90-day mortality comparing women and men. We also found in our multivariate analysis that there are no sex differences with respect to ICU length of stay (ICU LOS) and favorable CPC score at ICU discharge. These results were similar when this analysis was repeated in ICU survivors.

For our primary outcome, as discussed earlier, Mandigers et al. [15] reported a significant difference in 1-year mortality for OHCA patients between women and men. They also stated that one of their limitations was that no cardiac arrest characteristics could be included in their multivariable analysis. Our crude—unadjusted—analysis, i.e., Kaplan–Meier curve and associated log-rank test, showed a statistically significant difference (*p* = 0.029). As expected, when we included cardiac arrest characteristics, the sex difference is statistically insignificant (*p* > 0.05). These results are in line with previous studies [15,17,18].

When comparing descriptive statistics between women and men, women appear to have less favorable cardiac arrest characteristics, i.e., less initial shockable rhythms, less AED defibrillation use, and less frequent defibrillation by EMS. Women were less likely to have received interventions like PCI and CABG than men. Symptoms related to cardiac pathology can be differentially presented in women than in men [9]. This could make it more difficult to recognize cardiac problems in women, and can therefore lead to a delay in emergency response or effective treatment, although this is not clear from our data.

As for ICU length of stay, we found no difference between women and men. A study by Reinikainen et al. [19] concludes that the mean treatment time in ICU is higher for men then for women, even after adjusting for severity of illness. This contrasts our results, but their study focused on a general ICU population. Their results are in line with the hypothesis that men may be more susceptible to complications, such as sepsis, which in turn may influence ICU length of stay [20,21,22]. As for our analysis, we found no evidence for this in OHCA patients.

As for favorable CPC score at ICU discharge, we found no statistically significant difference between men and women. In a recent study, Karlsson and colleagues performed a similar analysis on 1667 patients from an international OHCA database and reported similar results in the multivariate analysis [22].

The results from the regression analyses contained several other notable results that we would like to discuss. First, we found that a home arrest compared to public arrest is significantly positively associated with the time to 90-day mortality. In other words, patients having a cardiac arrest at home have a higher 90-day mortality than people having an arrest in public. We speculate that public cardiac arrests may be reported more quickly and have quicker bystander BLS and emergency response.

Second, we report that AED defibrillation is associated with a lower time to 90-day mortality. At the same time, AED defibrillation is more frequently used in men (38.2%) than in women (29.5%, *p* = 0.006, Table 2). Both the incidence of shockable rhythms and the administration of AED shocks are less in women than in men, pointing towards a different pathophysiology. This is also reflected by our findings that women, compared to men, are less likely to have myocardial infarction as the underlying cause of the arrest, but more likely have primary rhythm or conduction problems.

Further, we found that women are less likely to have an initial shockable rhythm than men. The study by Blom et al. reported this as well [8]. They speculate that this may have to do with the difference in response to a cardiac arrest, in women compared to men, therefore delaying emergency response, or that women have a more rapid transition from a shockable rhythm to a non-shockable rhythm due to biological factors [8].

Finally, speculatively, a possible explanation as to why women are less likely to receive AED defibrillation is that bystanders are more hesitant to attach AED pads to a woman’s chest [23,24]. This is in line with a study from Kiyohara et al., who report that women (19–49 years) are less likely to have public AED pads applied than men from the same age group [25]. They hypothesize that, in a public setting, undressing a woman’s chest could be a problem for bystanders, resulting in less pad application. In addition, we found that AED defibrillation is positively associated with a favorable CPC score at ICU discharge. These results are in line with previous studies that showed that AED defibrillation use indeed has a positive impact on neurological outcome after OHCA [26,27].

### 4.1. Limitations

Our study has several limitations. First, initial rhythm was classified based on two different observations: either at the initial stage of the resuscitation process, or, if this was unknown, we based this on the first observation during transfer to the hospital, based on EMS. This may affect the classification of a primary rhythm being shockable or non-shockable. However, this is the best available proxy as properly documenting such information at the crucial moments during a cardiac arrest is difficult. Second, CPC scores at ICU discharge were retrospectively determined based on information gathered from patient charts and no CPC scores at longer-term follow-ups were available. Finally, the study period between 1 January, 2007 and 31 December, 2018, yielded a relatively large sample of patients but may have also been a possible confounder. Over this period, the cardiac care improved and that may have increased survival likelihood of CA patients. However, additional analysis, e.g., testing the effect of time on survival, may impact the sample size and in turn the power of the analysis.

### 4.2. Future Research

There are several possibilities for future research. First, we found that women were less likely to have a shockable rhythm than men. A question that arises is whether women indeed have a shockable rhythm less frequently as the primary cause of arrest, or, does it occur as frequently as in men, but other factors (i.e., rapid degeneration to non-shockable rhythm due to biological factors or delay in recognition) play a role. We recommend conducting these studies in larger retrospective multicenter (or even population-based) studies of all OHCA patients, including patients who die before ICU admission.

Second, CPC scores were retrospectively determined based on the patient charts. For future research, more objective measurements of CPC scores after hospital discharge should be used to serve as a better proxy for neurological outcome.

Third, future research should examine whether bystander hesitance (to undress a women’s chest before resuscitation) indeed contributes to a slower start and worse outcomes. Finally, future research should collect data (including clinically relevant cardiac arrest characteristics) in more hospitals, resulting in larger samples, spanning over a larger period of time.

## 5. Conclusions

We found that, when adjusting for cardiac arrest characteristics, there is no difference between women and men with respect to 90-day mortality, ICU LOS, and a favorable CPC score at ICU discharge. Our results contribute toward the knowledge regarding a differential underlying pathophysiology of cardiac arrest between both sexes; this should be studied in future research.

## Figures and Tables

**Figure 1 jcm-10-04286-f001:**
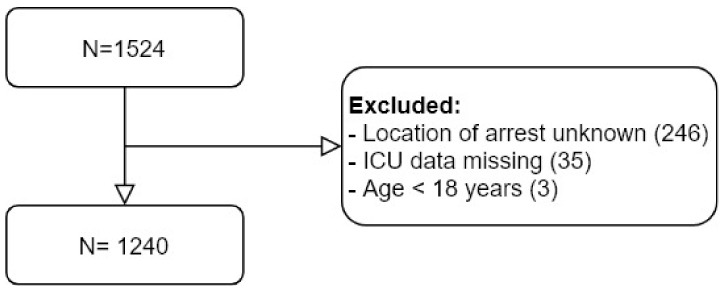
Flow diagram for included patients.

**Figure 2 jcm-10-04286-f002:**
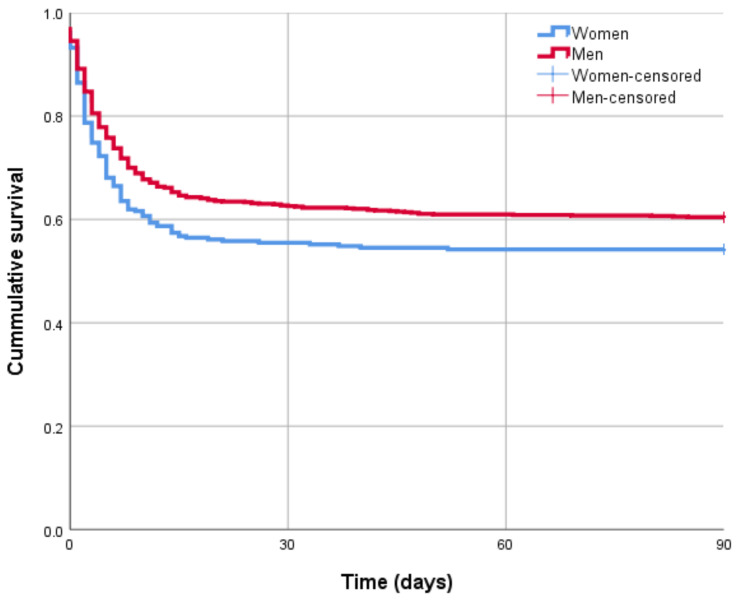
Kaplan–Meier curve (log-rank test *p*-value-0.029).

**Table 1 jcm-10-04286-t001:** Baseline characteristics for (non)-survivors.

	Total (N = 1240)	90-Day Mortality		*p*-Value	Missing Values (N)
		Non-Survivors (N = 510)	Survivors (N = 730)		
**Demographic characteristics**					
Age (years)	64.0 (53.8–73.0)	68.5 (59.7–77.4)	60.7 (50.5–69.3)	<**0.001**	0
Gender				0.053	0
Men	930 (75.0)	368 (72.2)	562 (77.0)		
Women	510 (25.0)	142 (27.8)	168 (23.0)		
**Medical history**					
Hypertension	435 (38.3)	187 (41.5)	248 (36.2)	0,074	104
Hypercholesterolemia	283 (25.0)	114 (25.5)	169 (24.7)	0.725	108
Diabetes Mellitus	229 (19.6)	136 (29.1)	93 (13.2)	<**0.001**	70
Family history of cardiovascular disease	199 (22.3)	42 (13.5)	157 (27.1)	<**0.001**	349
Previous myocardial infarction	279 (22.6)	116 (22.9)	163 (22.4)	0.815	5
Chronic heart failure	151 (12.2)	81 (16.1)	70 (9.6)	**0.001**	7
Previous cardiac arrhythmia	199 (17.0)	89 (19.2)	110 (15.5)	0.101	69
Previous TIA or stroke	105 (8.5)	60 (11.8)	45 (6.2)	<**0.001**	6
Peripheral vascular disease	111 (9.5)	64 (13.7)	47 (6.7)	<**0.001**	72
Smoking	345 (34.7)	134 (34.7)	211 (34.3)	0.736	246
Chronic lung disease	146 (11.8)	75 (14.8)	71 (9.8)	**0.007**	7
Pulmonary embolism	17 (1.4)	9 (1.8)	8 (1.1)	0.317	6
Previous percutaneous coronary intervention	166 (13.5)	64 (12.6)	102 (14.0)	0.479	8
Previous coronary artery bypass grafting	81 (6.6)	35 (6.9)	46 (6.3)	0.681	7
Previous ICD implantation	28 (2.3)	12 (2.4)	16 (2.2)	0.85	7
Previous pacemaker implantation	35 (2.8)	19 (3.7)	16 (2.2)	0.108	7
**Medication use history**					
Aspirin	233 (19.0)	104 (20.6)	129 (17.8)	0.222	11
Antiplatelet therapy	81 (6.6)	40 (7.9)	41 (5.7)	0.115	10
Oral anticoagulants	175 (14.2)	81 (16.0)	94 (13.0)	0.129	10
ACE inhibitors	221 (18.0)	90 (17.8)	131 (18.1)	0.912	10
Aldosterone antagonists	1 (0.1)	1 (0.1)	0 (0.0)	0.231	11
Angiotensin-II blocker	108 (8.8)	52 (10.3)	56 (7.7)	0.117	10
Beta blocker	303 (24.7)	127 (25.2)	176 (24.3)	0.712	11
Calcium channel blocker	98 (8.0)	50 (9.9)	48 (6.6)	0.037	10
Nitrates	85 (6.9)	37 (7.3)	48 (6.6)	0.631	10
Diuretics	235 (19.1)	114 (22.6)	121 (16.7)	0.01	10
Statins	302 (24.6)	133 (26.3)	169 (23.3)	0.225	10
Non-statin lipid lowering agents	19 (1.5)	7 (1.4)	12 (1.7)	0.707	10
Anti-arrhythmic	57 (4.6)	26 (5.1)	31 (4.3)	0.471	9
**Cardiac arrest characteristics**					
Location of arrest				<**0.001**	0
*Home*	654 (52.7)	317 (62.2)	337 (46.2)		
*Public*	586 (47.3)	193 (37.8)	393 (53.8)		
Witnessed arrest	908 (76.4)	340 (70.2)	566 (80.6)	<**0.001**	54
Bystander CPR	768 (64.9)	266 (55.1)	502 (71.7)	<**0.001**	57
Estimated time patient found until CPR				<**0.001**	234
*0–5 min*	777 (77,2)	263 (67.4)	514 (83.4)		
*6–10 min*	180 (17.9)	96 (24.6)	84 (13.6)		
*11–20 min*	41 (4.1)	26 (6.7)	15 (2.4)		
*> 20 min*	8 (0.8)	5 (1.3)	3 (0.5)		
Defibrillation by AED	442 (36.0)	110 (21.8)	332 (46.0)	<**0.001**	13
Initial rhythm				<**0.001**	65
*Shock*	924 (78.6)	298 (60.7)	626 (91.5)		
*Non-shock*	251 (21.4)	193 (39.3)	58 (8.5)		
Defibrillation by EMS	792 (64.1)	300 (59.2)	492 (67.6)	**0.002**	5
Time interval arrest to ROSC				<**0.001**	271
*0–5 min*	78 (8.0)	16 (4.0)	62 (10.9)		
*6–10 min*	191 (19.7)	48 (12.0)	143 (25.1)		
*11–20 min*	392 (40.5)	139 (34.8)	253 (44.5)		
*> 20 min*	308 (31.8)	197 (49.3)	111 (19.5)		
Intubation out of hospital	713 (57.9)	339 (67.4)	164 (32.6)	<**0.001**	8
Cause of arrest—no (%)				<**0.001**	74
*Cardiac cause*	1064 (91.3)	402 (85.2)	662 (95.4)		
*Non-cardiac cause*	102 (8.7)	70 (14.8)	32 (4.6)		
GCS at ED				<**0.001**	303
3 to 6	596 (63.6)	309 (88.8)	287 (48.7)		
7 to 11	145 (15.5)	25 (7.2)	120 (20.4)		
11 to 15	196 (20.9)	14 (4.0)	182 (30.9)		
Cardiac cause—no (%)				0.842	0
*Acute myocardial infarction*	681 (64.4)	258 (64.7)	423 (64.2)		
*Old myocardial infarction/scar*	121 (11.4)	47 (11.8)	74 (11.2)		
*Non-ischemic cardiomyopathy*	97 (9.2)	32 (8.0)	65 (9.9)		
*Primary rhythm or conduction disturbance*	108 (10.2)	40 (10.0)	68 (10.3)		
*Intoxication (heart medication)*	4 (0.4)	1 (0.3)	3 (0.5)		
*Cardiac tamponade*	1 (0.1)	0 (0.0)	1 (0.2)		
*Other*	15 (1.4)	6 (1.5)	9 (1.4)		
*Unknown cardiac cause*	31 (2.9)	15 (3.8)	16 (2.4)		
Non-cardiac causes—no (%)				**0.023**	0
*Epilepsy*	5 (5.0)	4 (5.8)	1 (3.1)		
*Trauma*	2 (2.0)	1 (1.4)	1 (3.1)		
*Intoxication*	10 (9.9)	6 (8.7)	4 (12.5)		
*Submersion*	6 (5.9)	5 (7.2)	1 (3.1)		
*Intracranial bleeding*	6 (5.9)	6 (8.7)	0 (0.0)		
*Pulmonary embolism*	18 (17.8)	11 (15.9)	7 (21.9)		
*Septic shock*	7 (6.9)	7 (10.1)	0 (0.0)		
*Asphyxia*	14 (13.9)	10 (14.5)	4 (12.5)		
*Hypoxia*	23 (22.8)	17 (24.6)	6 (18.8)		
*Other*	10 (9.9)	2 (2.9)	8 (25.0)		

Note: comparison of baseline characteristics at 90-day (non-)survivors after ICU admission. Numbers are median and interquartile range (IQR) or number and percentage for continuous or categorical variables, respectively; ED: emergency department, TIA: transient ischemic attack, ICD: implantable cardioverter defibrillator, ACE: angiotensin converting enzyme, CPR: cardiopulmonary resuscitation, AED: automatic external defibrillator, EMS: emergency medical service. GCS: Glasgow coma scale. Bold values are significant at 5% alpha level.

**Table 2 jcm-10-04286-t002:** Baseline characteristics for women and men.

	Total (N = 1240)	Women (N = 310)	Men (N = 930)	*p*-value	Missing Values (N)
**Demographic characteristics**					
Age (years)	64.0 (53.8–73.0)	61.8 (49.6–72.1)	64.4 (55.0–73.2)	**0.014**	0
**Medical history**					
Hypertension	435 (38.3)	117 (39.9)	318 (37.7)	0.503	104
Hypercholesterolemia	283 (25.0)	48 (16.7)	235 (27.8)	<**0.001**	108
Diabetes Mellitus	229 (19.6)	60 (20.2)	169 (19.4)	0.752	70
Family history of cardiovascular disease	199 (22.3)	46 (20.6)	153 (22.9)	0.48	349
Previous myocardial infarction	279 (22.6)	37 (12.0)	242 (26.1)	<**0.001**	5
Chronic heart failure	151 (12.2)	36 (11.7)	115 (12.4)	0.712	7
Previous cardiac arrhythmia	199 (17.0)	51 (17.3)	148 (16.9)	0.852	69
Previous TIA or stroke	105 (8.5)	22 (7.1)	83 (9.0)	0.312	6
Peripheral vascular disease	111 (9.5)	27 (9.3)	84 (9.6)	0.897	72
Smoking	345 (34.7)	73 (28.0)	272 (37.1)	**0.008**	246
Chronic lung disease	146 (11.8)	61 (19.7)	85 (9.2)	<**0.001**	7
Pulmonary embolism	17 (1.4)	9 (2.9)	8 (0.9)	**0.008**	6
Previous percutaneous coronary intervention	166 (13.5)	25 (8.1)	141 (15.3)	**0.001**	8
Previous coronary artery bypass grafting	81 (6.6)	9 (2.9)	72 (7.8)	**0.003**	7
Previous ICD implantation	28 (2.3)	8 (2.6)	20 (2.2)	0.657	7
Previous pacemaker implantation	35 (2.8)	12 (3.9)	23 (2.5)	0.201	7
**Medication use history**					
Aspirin	233 (19.0)	44 (14.3)	189 (20.5)	**0.016**	11
Antiplatelet therapy	81 (6.6)	13 (4.2)	68 (7.4)	0.053	10
Oral anticoagulants	175 (14.2)	52 (16.8)	123 (13.4)	0.13	10
ACE inhibitors	221 (18.0)	45 (14.6)	176 (19.1)	0.076	10
Aldosterone antagonists	1 (0.1)	0 (0.0)	1 (0.1)	0.563	11
Angiotensin-II blocker	108 (8.8)	29 (9.4)	79 (8.6)	0.649	10
Beta blocker	303 (24.7)	73 (23.8)	230 (24.9)	0.681	11
Calcium channel blocker	98 (8.0)	33 (10.7)	65 (7.0)	**0.04**	10
Nitrates	85 (6.9)	19 (6.2)	66 (7.2)	0.553	10
Diuretics	235 (19.1)	68 (22.1)	167 (18.1)	0.125	10
Statins	302 (24.6)	55 (17.9)	247 (26.8)	**0.002**	10
Non-statin lipids lowering agents	19 (1.5)	3 (1.0)	16 (1.7)	0.348	10
Anti-arrhythmic	57 (4.6)	14 (4.5)	43 (4.7)	0.935	9
**Cardiac arrest characteristics**					
Location of arrest				<**0.001**	0
*Home*	654 (52.7)	211 (68.1)	443 (47.6)		
*Public*	586 (47.3)	99 (31.9)	487 (52.4)		
Witnessed arrest	908 (76.4)	215 (74.4)	691 (77.0)	0.358	54
Bystander CPR	768 (64.9)	188 (64.8)	580 (64.9)	0.97	57
Estimated time patient found until CPR				0.535	234
*0–5 min*	777 (77.2)	185 (75.2)	592 (77.9)		
*6–10 min*	180 (17.9)	51 (20.7)	129 (17.0)		
*11–20 min*	41 (4.1)	8 (3.3)	33 (4.3)		
*> 20 min*	8 (0.8)	2 (0.8)	6 (0.8)		
Defibrillation by AED	442 (36.0)	90 (29.5)	352 (38.2)	**0.006**	13
Initial rhythm				<**0.001**	65
*Shock*	924 (78.6)	199 (67.2)	725 (82.5)		
*Non-shock*	251 (21.4)	97 (32.8)	154 (17.5)		
Defibrillation by EMS	792 (64.1)	175 (57.0)	617 (66.5)	**0.003**	5
Time interval arrest to ROSC				**0.803**	271
*0–5 min*	78 (8.0)	16 (6.6)	62 (8.5)		
*6–10 min*	191 (19.7)	47 (19.4)	144 (19.8)		
*11–20 min*	392 (40.5)	101 (41.7)	291 (40.0)		
*> 20 min*	308 (31.8)	78 (32.2)	230 (31.6)		
Intubation out of hospital	713 (57.9)	180 (58.8)	533 (57.6)	0.698	8
Cause of arrest—no (%)				<**0.001**	74
*Cardiac cause*	1064 (91.3)	234 (82.4)	830 (94.1)		
*Non-cardiac cause*	102 (8.7)	50 (17.6)	52 (5.9)		
GCS at ED				0.945	303
3 to 6	596 (63.6)	145 (62.8)	451 (63.9)		
7 to 11	145 (15.5)	36 (15.6)	109 (15.4)		
11 to 15	196 (20.9)	50 (21.6)	146 (20.7)		
Cardiac cause—no (%)				<**0.001**	0
*Acute myocardial infarction*	681 (64.4)	131 (56.2)	550 (66.7)		
*Old myocardial infarction*	121 (11.4)	19 (8.2)	102 (12.4)		
*Cardiomyopathy*	97 (9.2)	24 (10.3)	73 (8.8)		
*Primary rhythm or conduction disturbance*	108 (10.2)	41 (17.6)	67 (8.1)		
*Intoxication (heart medication)*	4 (0.4)	1 (0.4)	3 (0.4)		
*Cardiac tamponade*	1 (0.1)	1 (0.4)	0 (0.0)		
*Other*	15 (1.4)	4 (1.7)	11 (1.3)		
*Unknown cardiac cause*	31 (2.9)	12 (5.2)	19 (2.3)		
Non-cardiac causes—no (%)				0.451	0
*Epilepsy*	5 (5.0)	1 (2.0)	4 (7.8)		
*Trauma*	2 (2.0)	1 (2.0)	1 (2.0)		
*Intoxication*	10 (9.9)	5 (10.0)	5 (9.8)		
*Submersion*	6 (5.9)	3 (6.0)	3 (5.9)		
*Intracranial bleeding*	6 (5.9)	3 (6.0)	3 (5.9)		
*Pulmonary embolism*	18 (17.8)	13 (26.0)	5 (9.8)		
*Septic shock*	7 (6.9)	3 (6.0)	4 (7.8)		
*Asphyxia*	14 (13.9)	4 (8.0)	10 (19.6)		
*Hypoxia*	23 (22.8)	13 (26.0)	10 (19.6)		
*Other*	10 (9.9)	4 (8.0)	6 (11.8)		

Note: baseline characteristics comparing women and men. Numbers are median and interquartile range (IQR) or number and percentage for continuous or categorical variables, respectively; ED: emergency department, TIA: transient ischemic attack, ICD: implantable cardioverter defibrillator, ACE: angiotensin converting enzyme, CPR: cardiopulmonary resuscitation, AED: automatic external defibrillator, EMS: emergency medical service. GCS: Glasgow coma scale. Bold values are significant at 5% alpha level.

**Table 3 jcm-10-04286-t003:** Clinical outcomes for women and men.

	Total (N = 1240)	Women (N = 310)	Men (N = 930)	Missing Values (N)	*p*-Value
90-day mortality	510 (41.1)	142 (45.8)	368 (40.0)	0	0.053
ICU length of stay (days)	4.0 (2.0–7.0)	4.0 (2.0–7.0)	4.0 (3.0–7.0)	36	0.084
Favorable neurologic outcome, CPC 1–2	686 (55.3)	155 (50.0)	531 (57.0)	44	**0.038**
Poor neurologic outcome, CPC 3–5	510 (41.1)	142 (45.8)	368 (39.6)		

Note: Event is based on 90-day mortality after ICU admission. CPC is cerebral performance category. Favorable CPC is based on CPC score 1 and 2. Bold values are significant at 5% alpha level.

**Table 4 jcm-10-04286-t004:** Cox regression for the time to 90-day mortality.

		Cox Regression Analysis for the Time to 90-Day Mortality after ICU Admission
Men		1.150 (0.898–1.473)
Age		1.034 (1.025–1.042)
Medical history	Hypercholesterolemia	0.912 (0.710–1.170)
	Former myocardial infarction	1.064 (0.789–1.434)
	Chronic lung disease	0.905 (0.672–1.218)
	Pulmonary embolism	1.109 (0.564–2.180)
	Former PCI	0.782 (0.547–1.119)
	Former CABG	0.737 (0.480–1.132)
Medication use history	Aspirin	1.023 (0.775–1.350)
	Calcium channel blocker	0.896 (0.642–1.250)
	Statins	0.933 (0.706–1.233)
Cardiac arrest characteristics	Location of arrest	
	Home	**1.606 (1.293–1.996)**
	Public	Reference
	Defibrillation by AED	**0.540 (0.414–0.705)**
	Initial rhythm	
	Shock	**0.380 (0.287–0.505)**
	Non-shock	Reference
	Defibrillation by EMS	1.106 (0.828–1.477)
	Cause of arrest	
	Cardiac	**0.623 (0.439–0.884)**
	Non-cardiac	Reference
N		992
−2 LogLikelihood		5,003,168
Chi-square (df, *p*-value)		252.51 (16, <0.001)

Note: estimates are hazard ratios (HR) and 95% confidence interval (95% CI), derived from a Cox proportional hazard regression analysis. ICU: intensive care unit, PCI: percutaneous coronary intervention, CABG: coronary artery bypass graft, AED: automated external defibrillator, EMS: emergency medical service. Bold values are significant at 5% alpha level.

**Table 5 jcm-10-04286-t005:** Binary logistic and linear regression analyses results.

		Favorable CPC Score	ICU Length of Stay
		(model 1)	(model 2)
Men		1.123 (0.780–1.617)	0.428 (−0.442 to 1.298)
Age		0.957 (0.946–0.969)	−0.024 (−0.051 to 0.003)
Medical history	Hypercholesterolemia	0.876 (0.599–1.281)	0.017 (−0.893 to 0.927)
	Former myocardial infarction	1.065 (0.686–1.654)	0.659 (−0.430 to 1.749)
	Chronic lung disease	1.061 (0.650–1.733)	0.063 (−1.090 to 1.216)
	Pulmonary embolism	0.630 (0.196–2.031)	−0.537 (−3.398 to 2.325)
	Former PCI	1.520 (0.901–2.563)	−0.319 (−1.570 to 0.932)
	Former CABG	2.234 (1.167–4.280)	0.069 (−1.490 to 1.627)
Medication use history	Aspirin	0.898 (0.568–1.419)	−0.659 (−1.744 to 0.426)
	Calcium channel blocker	0.973 (0.568–1.419)	0.855 (−0.451 to 2.161)
	Statins	1.105 (0.715–1.708)	−0.227 (−1.268 to 0.814)
Cardiac arrest characteristics	Location of arrest		
	Home	**0.456 (0.336–0.619)**	Reference
	Public	Reference	0.056 (−0.675 to 0.787)
	Defibrillation by AED	**2.071 (1.457–2.945)**	0.259 (−0.585 to 1.104)
	Initial rhythm		
	Shock	Reference	0.030 (−1.126 to 1.185)
	Non-shock	**0.194 (0.118–0.317)**	Reference
	Defibrillation by EMS	0.747 (0.488–1.145)	0.648 (−0.309 to 1.606)
	Cause of arrest		
	Cardiac	**2.742 (1.376–5.465)**	Reference
	Non-cardiac	Reference	−0.570 (−2.130 to 0.991)
N		960	969
F-test (df; *p*-value)			0.882 (16,952; 0.590)
−2 LogLikelihood		1051.10	
Likelihood ratio test *p*-value		<0.001	
(Nagelkerke) R-square		0.303	0.015
AIC			

Note: estimates are odds ratio (OR), Beta, and 95% confidence interval in parentheses (95% CI), derived from a binary logistic regression analysis (model 1) and a linear regression analysis (model 2). CPC: cerebral performance score, ICU: intensive care unit, PCI: percutaneous coronary intervention, CABG: coronary artery bypass graft, AED: automated external defibrillator, EMS: emergency medical service. Bold values are significant at 5% alpha level.

## Data Availability

The data and analysis scripts are available upon request from the corresponding author.

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
