# Peer review of "Do Women Have a Higher Mortality Risk Than Men following ICU Admission after Out-of-Hospital Cardiac Arrest? A Retrospective Cohort Analysis"

_jcm, 2021, doi:10.3390/jcm10184286_

Round 1

Reviewer 1 Report

Well written presentation of the study conducted by the authors. Detailed descriptive statistics, good discusion section and updated reference list. 

Author Response

September 13th, 2021, Rotterdam, The Netherlands

Dear Editor of Journal of Clinical Medicine,

This letter accompanies our revised manuscript submitted to Journal of Clinical Medicine titled: “Do women have a higher mortality risk than men following an ICU-admission after an out-of-hospital cardiac arrest? A retrospective cohort analysis”. We will address the reviewer comments raised by the reviewer in the rebuttal letter next.

We hope to adhere to the reviewer comments and that you will consider our manuscript for publication in the Journal of Clinical Medicine.

Warm regards,

Wim Rietdijk, PhD

REBUTTAL LETTER TO REVIEWER 1

We thank the reviewer for his Reviewer 1 did not raise significant issues that we had to address, except for the some spelling and grammar mistakes. We went through the manuscript and have changes several small typo’s. We hope you are happy with our revision to your review.

Reviewer 2 Report

Good written paper, with clear methodological part.

2 major concerns:

  1. Why reducing to patients admitted to the ICU? Did you not introduce your own confounder?
  2. The length of the study period is another confounder?

Author Response

September 13th, 2021, Rotterdam, The Netherlands

Dear Editor of Journal of Clinical Medicine,

This letter accompanies our revised manuscript submitted to Journal of Clinical Medicine titled: “Do women have a higher mortality risk than men following an ICU-admission after an out-of-hospital cardiac arrest? A retrospective cohort analysis”. We will address the reviewer comments raised by the reviewer in the rebuttal letter next.

We hope to adhere to the reviewer comments and that you will consider our manuscript for publication in the Journal of Clinical Medicine.

Warm regards,

Wim Rietdijk, PhD

REBUTTAL LETTER TO REVIEWER 2

Reviewer two raised two major concerns that we would like to discuss. We will provide an rebuttal per reviewer point and how we adjusted the manuscript accordingly.

2 major concerns:

  1. Why reducing to patients admitted to the ICU? Did you not introduce your own confounder?

The aim of the present study is to specifically study patients that are critically ill and admitted at the ICU. Studying the outcomes of these patients are highly interesting because they survived the first episode of CA (namely, cardiopulmonary resuscitation). However, they are prone to haemodynamic deterioration/instability, ischaemia/reperfusion injury, and neurological damage (Mandigers et al., 2020). We added a sentence to the introduction where we explain the relevance for studying these critically ill patients. In addition, the study should be seen as an in-depth research given the reported higher 1-year mortality in ICU-admitted women in a previous nationwide Dutch study (Mandigers, J Crit Care 2021). Finally, we agree with the reviewer that ideally all OHCA patients should be investigated, including patients who die at the scene or before ICU admission. We added a sentence to the Discussion section / paragraph “Future research”.

  1. The length of the study period is another confounder?

We agree with the reviewer that the study period between January 1st, 2007 and December 31st, 2018 may be a confounder. On the one hand this yielded a relatively large sample of patients, on the other hand it may also be a possible confounder. Over this period the cardiac care improved and that may have increased survival likelihood of CA patients. However, additional analysis such as to test the effect of time on survival, may impact the sample size and in turn the power of the analysis.

For this reason, we have noted this now in the limitation section of the manuscript, but have not elaborated on it in our analysis.

REFERENCES

Mandigers, L., Termorshuizen, F., de Keizer, N. F., Gommers, D., dos Reis Miranda, D., Rietdijk, W. J., & Corstiaan, A. (2020). A nationwide overview of 1-year mortality in cardiac arrest patients admitted to intensive care units in the Netherlands between 2010 and 2016. Resuscitation147, 88-94.